# Ecological Safety Assessment and Analysis of Regional Spatiotemporal Differences Based on Earth Observation Satellite Data in Support of SDGs: The Case of the Huaihe River Basin

Shan Sang [1] , Taixia Wu [1], Shudong Wang [2],*, Yingying Yang [1], Yiyao Liu [1], Mengyao Li [1] and Yuting Zhao [1]

[1] School of Earth Sciences and Engineering, Hohai University, Nanjing 210098, China; sangshan@hhu.edu.cn (S.S.); wutx@hhu.edu.cn (T.W.); yyy@hhu.edu.cn (Y.Y.); liuyiyao@hhu.edu.cn (Y.L.); limy@hhu.edu.cn (M.L.); zhaoyt@hhu.edu.cn (Y.Z.)

[2] Aerospace Information Research Institute, Chinese Academy of Sciences, Beijing 100101, China

\* Correspondence: wangsd@radi.ac.cn

**Abstract:** Terrestrial ecosystems provide a variety of benefits for human life and production, and are a key link for achieving sustainable development goals (SDGs). The basin ecosystem is one type of terrestrial ecosystem. Ecological security (ES) assessments are an important component of the overall strategy to achieve regional sustainable development. The Huaihe River Basin (HRB) has the common characteristics of most basins, such as high population density, a rapidly developing economy, and many environmental problems. This study constructed an ES evaluation system by applying a pressure-state-response framework as an assessment method for the sustainable development of basins. Taking the HRB as an example, this study determined the ES status of the region from 2001 to 2019 and analyzed crucial factors for any variation observed by combining remote sensing and climate data, relevant policies, and spatial information technology. The results highlight the importance of reserves and the negative impact of urban expansion on ES. Additionally, the enactment of policies had a positive impact on ES, whereas precipitation had a negative effect on ES in most areas of the HRB. Based on these results, the government should strengthen the protection of forests, grasslands, and wetlands and improve water conservation facilities. This study provides guidance for the subsequent economic development, environmental protection, and the achievements of SDG 15 in the HRB.

**Keywords:** SDGs; ecological safety; pressure-state-response; remote sensing; Huaihe River Basin

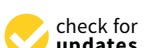

## 1. Introduction

The 193 member states of the United Nations adopted the 2030 Agenda for Sustainable Development and its 17 Sustainable Development Goals (SDGs) in August 2015 [1]. The 17 SDGs aim to protect the Earth and ensuring peace and prosperity for humankind [2], and provide all countries with a common approach and an agenda to address the serious challenges facing the world today, such as poverty, climate change, and conflicts [3]. SDG 15 is focused on the terrestrial ecosystem, which is the basis of human development, and provides various benefits for human production and life. The aim of SDG 15 is to protect, restore, and promote sustainable use of land; sustainably manage forests; combat desertification; halt land degradation; and prevent biodiversity loss [4].

The basin ecosystem is considered an important component of the terrestrial ecosystem. It is a reasonably closed system with clear boundaries that are involved with material and energy exchange with the external environment [5]. The basin ecosystem provides multiple ecosystem goods and services to residents, including freshwater and oxygen, supplies, purified water, and carbon fixation [6]. With rapid urbanization, extensive economic development, and explosive population growth, basins currently face several environmental problems, such as

water pollution, reduced species diversity, and ecosystem degradation [7]. As a vital part of the terrestrial ecosystem, the protection and restoration of basin ecosystems are crucial to achieving SDG 15. Existing studies on SDG 15 have quantitatively monitored a specific indicator, such as land degradation [8], forest management [9], or vegetation in mountainous areas [10]. There are a few articles on the comprehensive evaluation model of the SDG 15.

Ecological security (ES) assessments play an important role in determining the improvement needs of a specific basin and the sustainable development capacity of the region [11]. ES broadly refers to the maintenance of survival and development involving humans and other organisms; it includes natural, economic, and socioecological security [12]. ES evaluation results are used to analyze the effects of processes related to human development and construction activities on the environment over a specific period of time [13], to support the achievement of SDG 15. Many models for evaluating ES have been established, such as the pressure-state-response (PSR) [14], driver-state-response [15], driver-pressure-state-impact-response (DPSIR) [16], and ecosystem-based-management-driver-pressure-state-exposure-response models [17]. For example, Tang, et al. [18] analyzed the effects of implementing the River Chief System using an ES evaluation system based on the PSR framework. Hua et al. [19] developed an ES indicator system to assess the Dali Bai Autonomous Prefecture based on the PSR model. They showed that the region was in a good environmental condition, in terms of ES status. Shao et al. [20] adopted the DPSIR model to assess the state of coastal ecological environmental security.

The PSR model was jointly proposed by the Organization for Economic Cooperation and Development and the United Nations Environment Program in the late 1980s [21]. It classifies ecological indicators according to interactions between humans and ecosystems, as well as factors influencing such interactions, thereby allowing systematic evaluations [22]. The pressure is the ecological stress, reflecting direct pressures faced by a natural ecosystem from natural disasters and human socioeconomic activities. The state refers to the current health status of the ecosystem. The response refers to the policies, regulations, and other measures taken by human beings to ensure ecological sustainability in the context of ecological degradation [18]. The PSR identifies clear causal relationships and emphasizes the interactions between human activities and the environment [22]. It is applied most widely to evaluate the ES in different regions [23], such as the Tibetan Plateau [24], the Shiyang River Basin [25], and Kunming [26]. Because the PSR model is highly systematic and operable, this study selected an appropriate indicator system after considering the actual study area and the difficulty acquiring data. Therefore, the PSR was chosen as the basic framework for constructing the ES indicator system. The analytic hierarchy process (AHP) is a multi-objective decision analysis method that combines qualitative and quantitative criteria [27], and was introduced by Saaty [28] first and widely used to determine weights in ES studies. For example, Zheng and Li [29] used the AHP to determine the weights to set up the ES indicator system for a mineral-resource enterprise. Gao et al. [30] constructed an indicator system using the AHP to assess the ES of Pingtan Island.

With the development of remote sensing (RS) and geographic information system (GIS) technologies, ES assessment has evolved from analyses of single time series to visual analyses of spatiotemporal changes. The development of spatial information technologies provides abundant data for studying ES and enables more obvious and accurate spatiotemporal characteristics. For instance, Hazbavi, et al. [31] constructed an ES evaluation system to dynamically monitor the health of the Shazand watershed using Landsat5 TM and Landsat 8 OLI data. Wei et al. [32] used HJ1 satellite Charge Coupled Device images to determine the distribution of the landscape ES pattern for Ganzhou.

The Huaihe River Basin (HRB) is characterized by high population density, rapid economic development, a high land utilization rate, and numerous ecological problems [33]. These features are the same as most basins, so it is well suited as a typical case. This study aimed to: (1) construct an ES evaluation system applicable to most basins based on the PSR framework and AHP method, and provide an assessment method for the sustainable development of basins; (2) identify the driving factors affecting sustainable development of the basin; and (3) offer guidance for the realization of SDG 15 in the HRB. Considering the focus of SDG

15, and combined with literature on the actual HRB situation, this study selected indicators based on the PSR framework. Then, the weights of the indicators were determined using the AHP and an ES indicator assessment system was established. By combining of satellite data and GIS technology, this study revealed the spatiotemporal variation characteristics of ES in the basin. Finally, this study analyzed the spatiotemporal changes in ES and explained the changes using climate and policy data. The results will be useful to determine the causes of eco-environment degradation and provide guidance for implementing SDG 15.

## 2. Materials and Methods

### 2.1. Study Area

The HRB is located in the eastern part of China and covers an area of approximately $2.7 \times 10^5$ km$^2$ crossing Jiangsu, Anhui, Shandong, and Henan Provinces [34]. The main stream of the Huaihe River originates from Tongbai Mountain in Henan and flows into the sea at Yangzhou and Yancheng in Jiangsu [35]. There are many tributaries in the middle and upper reaches of the Huaihe River. The Dabie Mountains and hills in the Jiang-Huaihe region are the sources of the southern tributaries. Tributaries are typically characterized by a rushing current and are close to the source [36]. Plains comprise the main landform type in the basin, and the elevation in the western region is higher than the elevation in the eastern region (Figure 1a). The basin is situated in a climate transition zone, with a humid southern region and a semi-humid northern region [35]. The mean annual precipitation and temperature are approximately 883 mm and 11–16 °C, respectively [37].

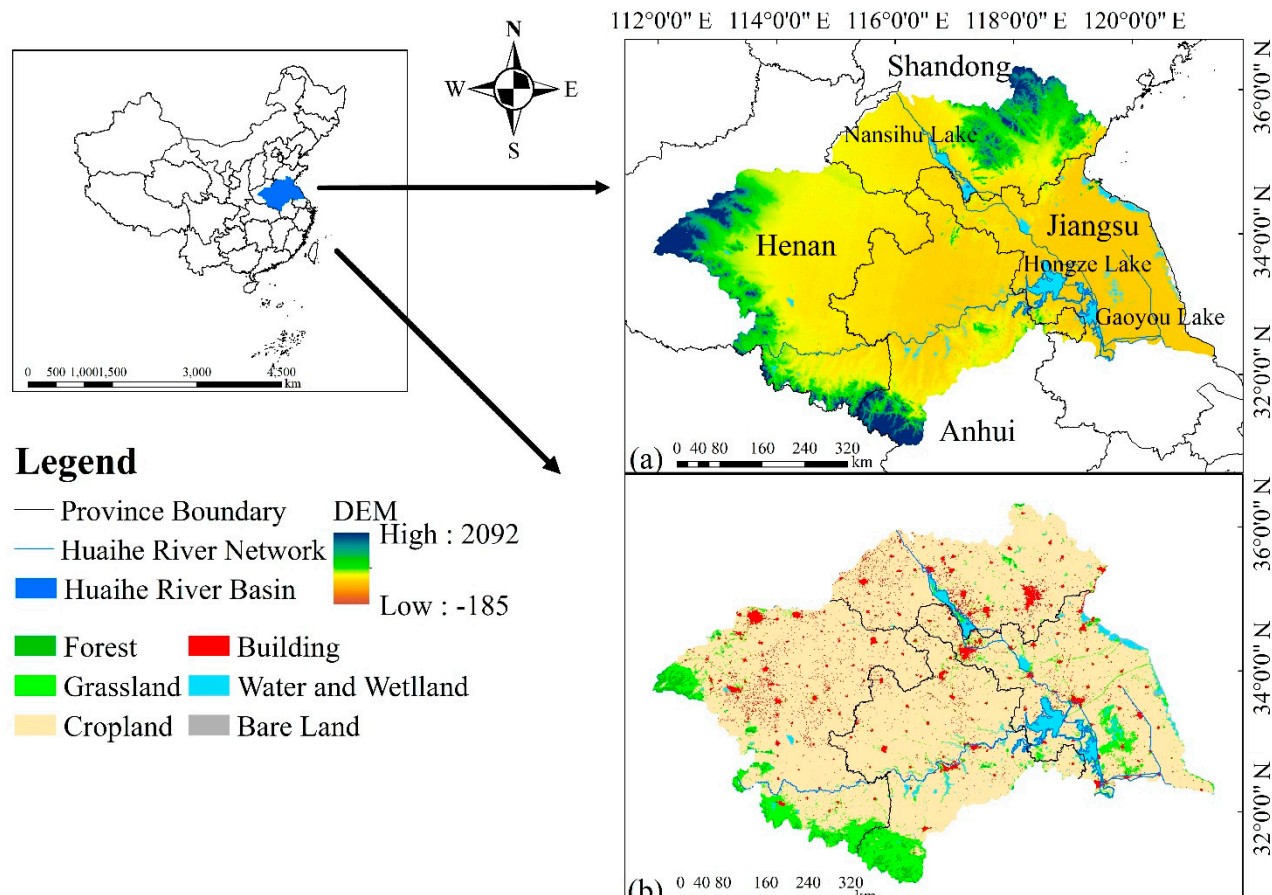

**Figure 1.** Location of the HRB. (**a**) Digital elevation model-derived map (from ASTER GDEM data) of the HRB showing Jiangsu, Anhui, Shandong, and Henan provinces and three main lakes (Nansihu Lake, Hongze Lake, and Gaoyou Lake). (**b**) Land use map (from MOD12Q1 data) of the HRB in 2001.

The HRB is densely populated with 611 people/km² [38] and has experienced a high rate of population growth in recent decades. The utilization rate of land is very high; cropland accounts for the largest proportion of the total area (Figure 1b). The HRB is an important grain, cotton, and oilseed production region in China [39], resulting in high land utilization. The industries mainly consist of coal, power, and light textile industries, which have developed rapidly in recent years. Hence, several industrial cities (e.g., Zhengzhou, Xuzhou, and Lianyungang) have emerged.

### 2.2. Workflow

The study selected indicators from ecological conditions and the literature, and determined the weights of the indicators using the AHP and expert judgment. After constructing the indicator evaluation system, the value of each indicator was calculated using satellite, statistical, and geospatial data. Then, the spatial distribution of ES was obtained in each year from 2001 to 2019 by normalizing the results. Finally, combining the precipitation, land use, and policy information, the spatiotemporal variations of ES were analyzed (Figure 2).

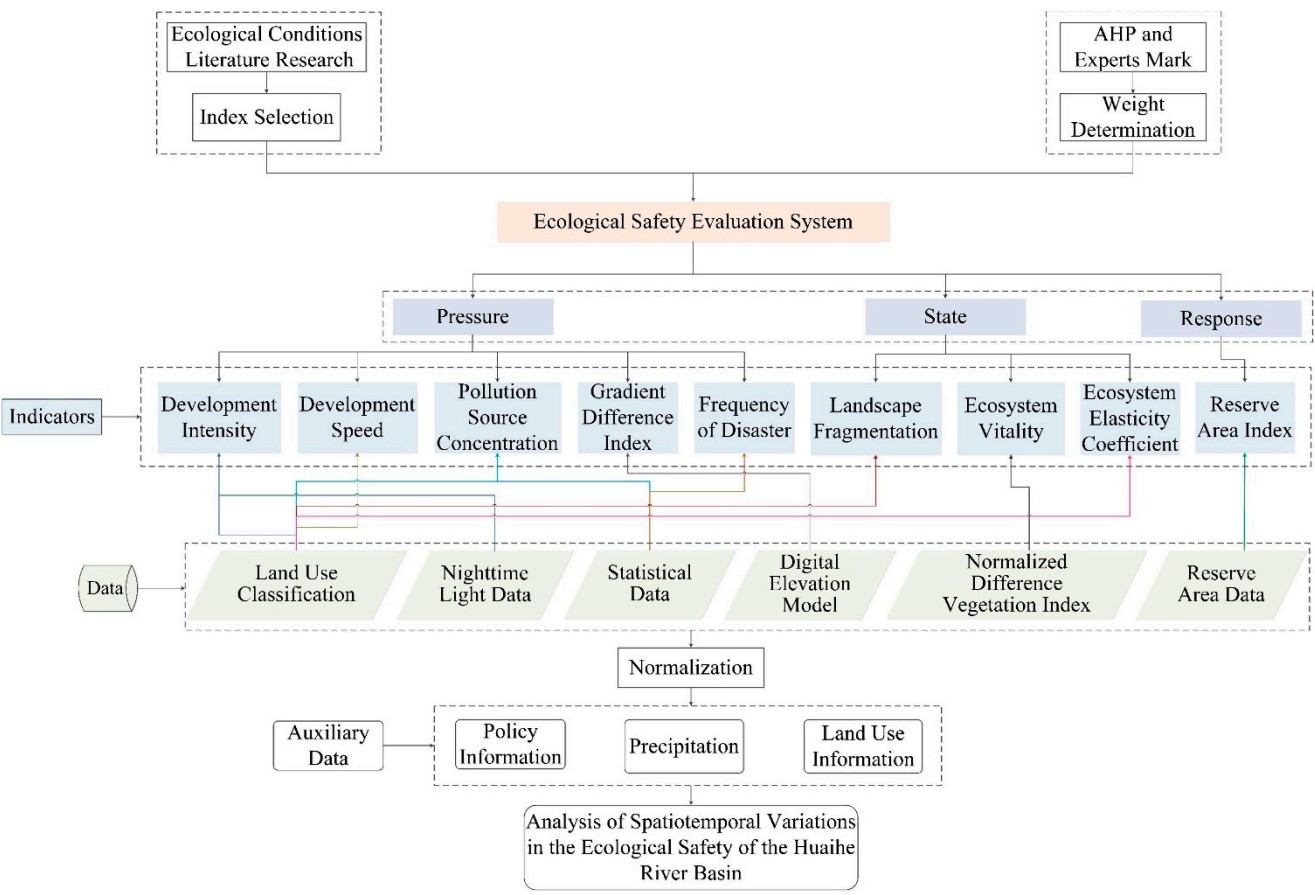

**Figure 2.** The flowchart of ES spatiotemporal original difference analysis for the HRB.

### 2.3. Establishment of the ES Assessment System

2.3.1. Indicator Calculation

(1) Development intensity refers to the intensity of human construction activities. The overall economy of the HRB has been growing, but economic development does not always occur in a balanced manner [40]; excessive development and urbanization have harmed the environment [41].

Development intensity was calculated as follows:

$$DI = B/S_G \times PCC, \tag{1}$$

$$PCC = (L_{ave} - L_{min})/(L_{max} - L_{min}), \tag{2}$$

where $DI$ is the development intensity in the grid, the size of the grids used in the study was 1 km $\times$ 1 km, $B$ is the built-up area in the grid, $S_G$ is the grid area, $PCC$ is the population concentration coefficient of the grid, $L_{ave}$ is the average light brightness in the grid, $L_{min}$ is the minimum light brightness in the grid, and $L_{max}$ is the maximum light brightness in the grid. The formula was improved based on the method proposed by Li et al. [42].

(2) The development speed is the area of building land expansion in the region each year. Rapid economic development in the HRB has resulted in damage to the eco-environment [41].

The development speed was calculated as follows:

$$DS = B_y - B_{y-1}, \tag{3}$$

where $DS$ is the development speed in the grid, $B_y$ is the developed area in the grid in year $y$, and $B_{y-1}$ is the developed area in the grid in year $y-1$.

(3) The pollution load is the total amount of point-source pollution from industries and urban residential areas and non-point-source pollution from pesticides and fertilizers into the water body in 1 year [43]; water pollution in the HRB is very serious [41].

The pollution load was calculated as follows:

$$PL = \sum_{i=1}^{6} E_i \times C_i + PV, \tag{4}$$

where $PL$ is the pollutant source concentration in the grid, $E_i$ is the output coefficient of the pollutant for land use type $i$, $C_i$ is the area of land use type $i$, and $PV$ is the pollutant point-source concentration. The validity of the calculation method and $C_i$ were confirmed by Chen et al. [44].

(4) The gradient difference is caused by the topography, as significant regional differences in elevation affects the precipitation, temperature, and other natural factors [45]; they have substantial impacts on human socioeconomic activities.

The gradient difference index was calculated as follows:

$$GDI = G_{var}/G_{ave}, \tag{5}$$

where $GDI$ is the gradient difference index for the grid, $G_{var}$ is the gradient variance in the grid, and $G_{ave}$ is the average gradient in the grid. The gradient was calculated using the DEM data.

(5) The frequency of disasters refers to the number of disasters occurring in 1 year. Drought and flood disasters in the HRB strongly influence the local socioeconomic landscape and environment [46].

The frequency of disasters was calculated as follows:

$$FOD = T_y, \tag{6}$$

where $FOD$ is the frequency of disasters occurring in the basin and $T_y$ is the number of disasters that occurred in year y.

(6) Landscape fragmentation, in which habitats or areas of specific land use types are broken down into smaller plots, represents human disturbance of the landscape [47]. The extent of fragmentation can be preliminarily assessed by the number of patches in a particular landscape [47].

Landscape fragmentation was calculated as follows [42]:

$$LF = \sum_{I=1}^{6} N_i/S_G, \tag{7}$$

where $LF$ is the extent of landscape fragmentation in the grid, $N_i$ is the number of land patches in the grid, and $S_G$ is the grid area.

(7) Ecological vitality refers to the integrity and health of the ecosystem supporting its ability to flourish and develop [48]. The quality of vegetation growth is a sign of ecosystem vitality [49].

Ecosystem vitality was calculated as follows:

$$EV = NA_y,\qquad(8)$$

where $EV$ is the ecosystem vitality of the grid, and $NA_y$ is the average NDVI in the grid.

(8) Ecological elasticity refers to the ability of an ecosystem to maintain and adjust itself, while resisting various external pressures and disturbances [50].

Ecosystem elasticity was calculated as follows:

$$EEC = \sum_{i=1}^{6} f_i \times C_i / S_G,\qquad(9)$$

where $EEC$ is the ecosystem elasticity for the grid, $f_i$ is the ecosystem elasticity weight for land use $i$, $C_i$ is the area of land use type $i$, and $S_G$ is the grid area. The validity of the formula was confirmed by Li, Sun, Zhu, and Cao [42], and the ecosystem elasticity weight for each land use type was determined based on expert consultation as follows: forest, 1; grassland, 0.8; cropland, 0.6; building, 0.2; water and wetland, 1; and bare land, 0.4.

(9) Reserves are established to conserve biodiversity and prevent habitat fragmentation. The area of reserves in the region indicates the importance of ecological protection by the local government [51].

The reserve area index was calculated as follows:

$$RI = S_R / S_G,\qquad(10)$$

where $RI$ is the reserve area index of the grid, $S_R$ is the area of reserve in the grid, and $S_G$ is the grid area.

### 2.3.2. Determination of Weights

This study combined the AHP and the expert's judgment to determine the weights of the indicators (Table 1). The AHP divides complex problems into a target layer, a criterion layer, and a parameter layer according to the relationship between internal factors, and forms a judgment matrix according to the importance of each factor in the same layer. The maximum eigenvector and random consistency ratio (CR) of the judgment matrix were calculated. The judgment matrix is effective when CR < 0.1. The weight is obtained by normalizing the maximum eigenvector [52].

**Table 1.** Indicators used in the ES assessment system.

| Target Layer | Criterion Layer (Weight) | Parameter Layer (Weight ±) |
|---|---|---|
| Ecological safety (ES) | Pressure (P) (0.3) | Development intensity (0.2−) |
| | | Development speed (0.2−) |
| | | Pollution load (0.3−) |
| | | Gradient difference index (0.1−) |
| | | Frequency of disaster (0.2−) |
| | State (S) (0.5) | Landscape fragmentation (0.3−) |
| | | Ecosystem vitality (0.3+) |
| | | Ecosystem elasticity (0.4+) |
| | Response (R) (0.2) | Reserve area index (1+) |

The CR of the judgement matrix in the pressure layer was 0.0022, the CR of the judgement matrix in the state layer was 0.0462, and the CR of the judgement matrix in the target layer was 0.0079. All weights were obtained when CR values met the requirements.

The indicators were divided into two types: positive (+) and negative (−) depending on their effect on the environment.

The ES value was calculated using the following formula:

$$V_{ES} = 0.3 \times V_p + 0.5 \times V_S + 0.2 \times V_R, \tag{11}$$

where $V_{ES}$ is the ES value, $V_P$ is the pressure value, $V_S$ is the state value, and $V_R$ is the response value. The range of possible ES values (0–1) can be divided into five equal intervals, such that each interval represents a different ES level (Table 2).

**Table 2.** Classification of ES assessment results.

| Value | Level | Description |
|:---:|:---:|:---:|
| 0~0.2 | 1 | Lower ES, worse sustainable development ability |
| 0.2~0.4 | 2 | Low ES, poor sustainable development ability |
| 0.4~0.6 | 3 | Medium ES, medium sustainable development ability |
| 0.6~0.8 | 4 | High ES, good sustainable development ability |
| 0.8~1 | 5 | Higher ES, better sustainable development ability |

### 2.4. Data Sources and Processing

The data used in this study were mainly satellite data. They were processed using GIS and RS-related software (e.g., ArcGIS and ENVI); data analysis was carried out using MATLAB software. All satellite data were processed by mosaic, clipping, projection, and format conversion, including nighttime light data, precipitation, surface temperature, digital elevation model (DEM), and MODIS product (i.e., MODIS/Terra Vegetation Indices 16-Day L3 Global 250 m SIN Grid (MOD13Q1 NDVI) and MODIS/Terra + Aqua Land Cover Type Yearly L3 Global 500 m SIN Grid (MCD12Q1 Land Cover Type)). The MOD13Q1 NDVI data was processed by S-G filtering and the maximum value composites (MVC) to eliminate the influence of noise and clouds. Nighttime light data were obtained from the Defense Meteorological Satellite Program Operational Linescan System and the National Polar Partnership's Visible Infrared Imaging Radiometer Suite (NPP/VIIRS). NPP/VIIRS data were derived from the Flint Earth Luminous product, which was provided by the Chinese Academy of Sciences and removed abnormal values. The nighttime light data also underwent supersaturation correction and continuity correction, in accordance with the methods used by Cao et al. [53] and Liang et al. [54]. Considering the difference in the spatial resolution of the data, the regional statistics tool in ArcGIS was used to calculate the mean value of each indicator data in $1 \times 1$ km grids and assigned the value to grids for subsequent normalization and the weighted calculation. The nighttime light data was used for calculating the development intensity. The MOD13Q1 NDVI data was used to calculate the ecosystem vitality. The DEM data was used for calculating the gradient difference index. The reserve area data was used to calculate the reserve area index. The statistical data was used for calculating the pollution load and frequency of disaster. The MCD12Q1 Land Cover Type was used to calculate the development intensity, development speed, pollution load, landscape fragmentation, and ecosystem elasticity.

#### 2.4.1. Data Sources

This study combined satellite data from different sources and vector data to determine the ES status of the HRB, whereas climate data and policy information were used to analyze variations in the ES (Table 3).

**Table 3.** Data source information.

| Data Type | Attributes | Acquisition Dates | Sources |
|---|---|---|---|
| Nighttime light data | satellite data | 2001–2019 | https://www.ngdc.noaa.gov/eog/dmsp/downloadV4composites.html http://satsee.radi.ac.cn/cfimage/nightlight/ (accessed on 2 January 2021) |
| Precipitation | satellite data | 2001–2019 | National Earth System Science Data Center |
| Surface temperature | satellite data | 2001–2019 | (http://www.geodata.cn (accessed on 10 November 2020)) |
| Digital elevation model (DEM) | satellite data | - | Geospatial Data Cloud (http://www.gscloud.cn/ (accessed on 10 November 2020)) |
| MOD13Q1 NDVI | satellite data | 2001–2019 | NASA LAADS DAAC (https://ladsweb.modaps.eosdis.nasa.gov/search (accessed on 30 November 2020)) |
| MCD12Q1Land Cover Type | satellite data | 2001–2019 | NASA LAADS DAAC (https://ladsweb.modaps.eosdis.nasa.gov/search (accessed on 30 November 2020)) |
| Basin boundary | geospatial data | - | Resource and Environment Data Cloud Platform (http://www.resdc.cn/Default.aspx (accessed on 1 November 2020)) |
| Country boundary | geospatial data | - | Resource and Environment Data Cloud Platform (http://www.resdc.cn/Default.aspx (accessed on 10 November 2020)) |
| Reserve area | geospatial data | - | World Database on Protected Areas (https://www.protectedplanet.net/?from=singlemessage&isappinstalled=0 (accessed on 10 November 2020)) |
| Statistical data | other | 2001–2019 | Information Center of the Ministry of Water Resources of Chinahttp://xxzx.mwr.gov.cn/xxgk/gbjb/sqnb/ (accessed on 10 November 2020) Hydrology Bureau of the Huaihe River Water Conservancy Committee http://www.hrc.gov.cn/swj/szygb/24350.jhtml (accessed on 10 December 2020) |
| Policy information | other | 2001–2019 | The People's Government of Jiangsu Province (http://en.jiangsu.gov.cn/ (accessed on 10 March 2021)) The People's Government of Anhui Province (http://english.ah.gov.cn/ (accessed on 10 March 2021)) The People's Government of Henan Province (http://english.henan.gov.cn/ (accessed on 10 March 2021)) The People's Government of Shandong Province (http://www.shandong.gov.cn/index.html (accessed on 10 March 2021)) |

### 2.4.2. Data Normalization

To eliminate differences (e.g., in terms of magnitude and units) among the indicators, the indicator values were transformed, such that all indicators were dimensionless with values that ranged between 0 and 1 [55]. The transformation formulas used were as follows:

$$X_s = \frac{x - min(x)}{max(x) - min(x)}, \tag{12}$$

$$X'_s = 1 - \frac{x - min(x)}{max(x) - min(x)}, \tag{13}$$

where $x$ is the original value; $max(x)$ and $min(x)$ are the maximum and minimum values of indicator $x$, respectively; $X_s$ is the standard value of a positive indicator; and $X'_s$ is the standard value of a negative indicator.

### 2.4.3. Land Change Analysis

The land-use transfer matrix is the main method used for investigating conversion among different land-use types and is written mathematically as follows [56]:

$$
P_{ij} = \begin{bmatrix} P_{11} & P_{12} & \cdots & P_{1n} \\ P_{21} & P_{22} & \cdots & P_{2n} \\ \vdots & \vdots & \vdots & \vdots \\ P_{n1} & P_{n2} & \cdots & P_{nn} \end{bmatrix},
\tag{14}
$$

where $P$ represents the area of interest; $n$ represents the number of land-use types; and $i$ and $j$ represent the land use types at the beginning and end of the study period, respectively.

### 2.4.4. Trend Analysis

Sen's slope estimator $Q_i$ can be used to assess trends in time series. This method was used to obtain the trends of changes in ES and annual average precipitation. $Q_i$ was calculated as follows [57]:

$$
N = n(n+1)/2,
\tag{15}
$$

$$
Q_i = \frac{x_j - x_k}{j - k} \ (for \ \ i = 1, 2, \ldots, N),
\tag{16}
$$

$$
Q_{med} = \begin{cases} Q_{(N+1)/2} & N \ is \ odd \\ \frac{1}{2}(Q_{N/2} + Q_{N+2/2}) & N \ is \ even \end{cases},
\tag{17}
$$

where $n$ is the number of datapoints in the time series; $x_j$ and $x_k$ are the data observed in year $j$ and $k$ (j > k), respectively; and $Q_{med}$ is the median of $Q_i$.

### 2.4.5. Correlation Analysis

The Pearson's correlation coefficient is used to measure the strength of the correlation between two variables. Here, we calculated correlation coefficients using the following formula to assess the correlation between precipitation and the degree of ES [58]:

$$
r_{x,y} = \frac{\sum_{i=1}^{n}(x_i - x)(y_i - y)}{\sqrt{\sum_{i=1}^{n}(x_i - x)^2}\sqrt{\sum_{i=1}^{n}(y_i - y)^2}},
\tag{18}
$$

where $n$ is the number of time series datapoints; $x_i$ and $y_i$ are the precipitation amount and ES value, respectively; $x$ and $y$ are the average precipitation amount and ES value, respectively; and $r_{x,y}$ is the correlation coefficient.

## 3. ES Assessment Results for the HRB

### 3.1. Overall Spatiotemporal Variations in ES of the HRB

To more clearly observe the variations in the ES of the HRB, 1 km × 1 km grids were set up within the HRB area. Each grid contained the result of indicator normalization and weighting, which facilitated intuitive visualization of spatial variation across the region (Figure 3). Then, proportions of areas with different levels of ES in the HRB were calculated (Figure 4).

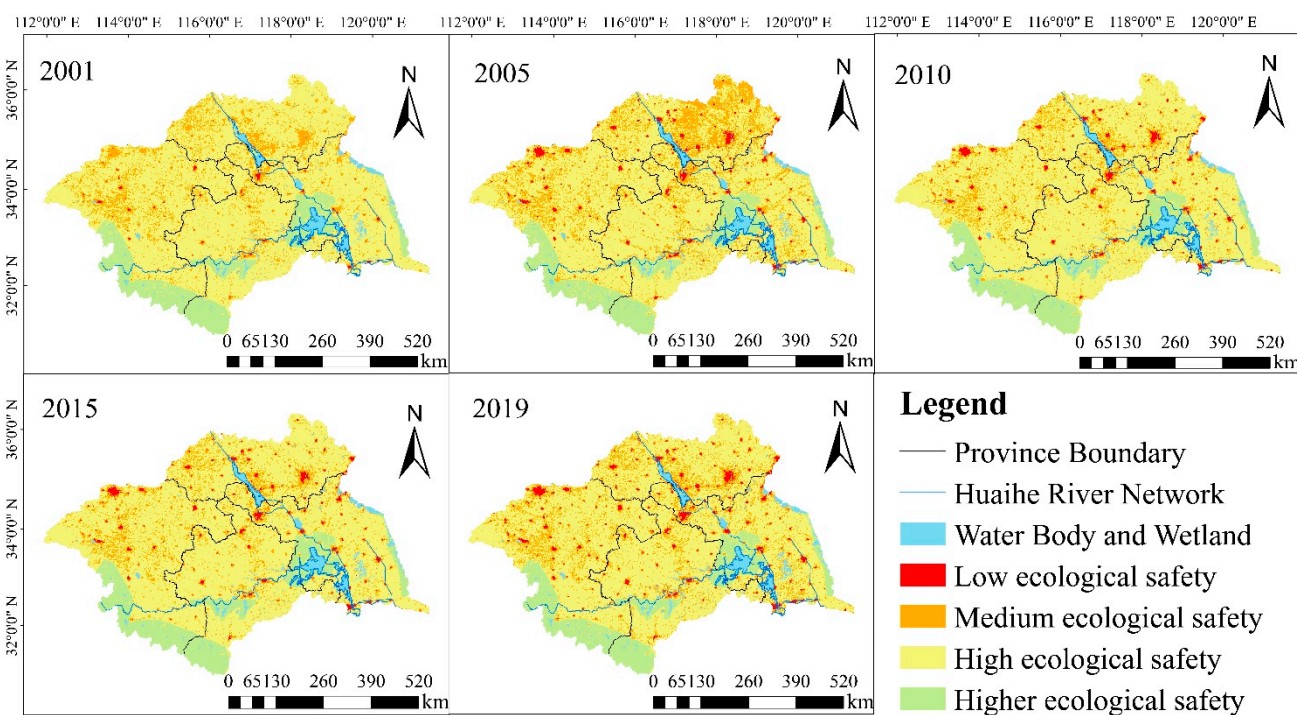

**Figure 3.** Spatial distribution of ES levels in the HRB in 2001, 2005, 2010, 2015, and 2019.

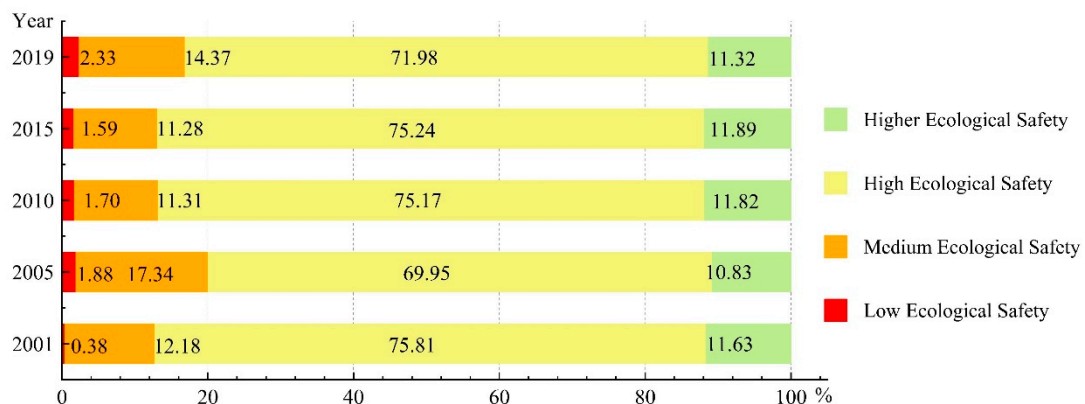

**Figure 4.** Proportions of areas with different ES levels in the HRB.

Areas with a low ES level were rare in the HRB, and most areas had a high ES level. Areas of low ES were mainly distributed over built-up areas, whereas areas of medium ES were generally scattered and located near built-up areas. Areas of higher ES were located in the southwestern and eastern parts of the basin, as well as a southeastern area near Hongze Lake. These areas coincide with the distribution of reserves and comprise grassland and forest areas. Over the 2001–2019 period, the spatial distribution of ES levels did not change greatly. In 2005, the total area with a medium ES level appeared to increase, mainly caused by the degradation of areas of high ES. The degraded regions were mostly located in the northeastern and western parts of the basin. In 2010, the areas where the ES rating decreased in 2005 exhibited improvement, such that they had a medium ES level. In 2015, there was little change in the spatial distribution of ES levels from 2010. In 2019, areas in Shandong and Henan provinces exhibited obvious changes in their ES ratings, whereas no clear changes were observed in the other two provinces.

The proportion of area with low ES remained low and did not extensively change during the 2001–2019 period. The maximum value of 2.33% was reached in 2019. The maximum proportion of area with medium ES (17.34%) was observed in 2005, and the

proportion decreased to its minimum value (11.31%) in 2010. A large proportion of the basin had a high ES level each year. In 2001, the proportion of area with high ES reached its maximum value of 75.81%. After that, it decreased sharply to its minimum value of 69.95% in 2005 (decrease of 5.86%). Subsequently, it increased again in 2010 and 2015, then decreased to 71.98% in 2019. From 2001 to 2015, the proportion of area with higher ES first decreased, then increased. The minimum value of 10.83% was reached in 2005, and the peak value of 11.89% was attained in 2015. The proportion then decreased slightly by 0.57% in 2019.

In summary, most areas of the HRB had good sustainable development ability. The better and poor sustainable development ability were associated with reserves and built-up areas, respectively. Shandong had a comparatively worse sustainable development ability, relative to the other provinces. Using 2001 as the starting point, the HRB exhibited its worst sustainable development ability in 2005 and its best sustainable development ability in 2015.

### 3.2. Variations of ES in the Four Provinces

The HRB area was divided into four province areas (Jiangsu, Anhui, Shandong, and Henan) based on provincial boundaries. The proportions of areas with different ES levels in each province were then calculated (Figure 5).

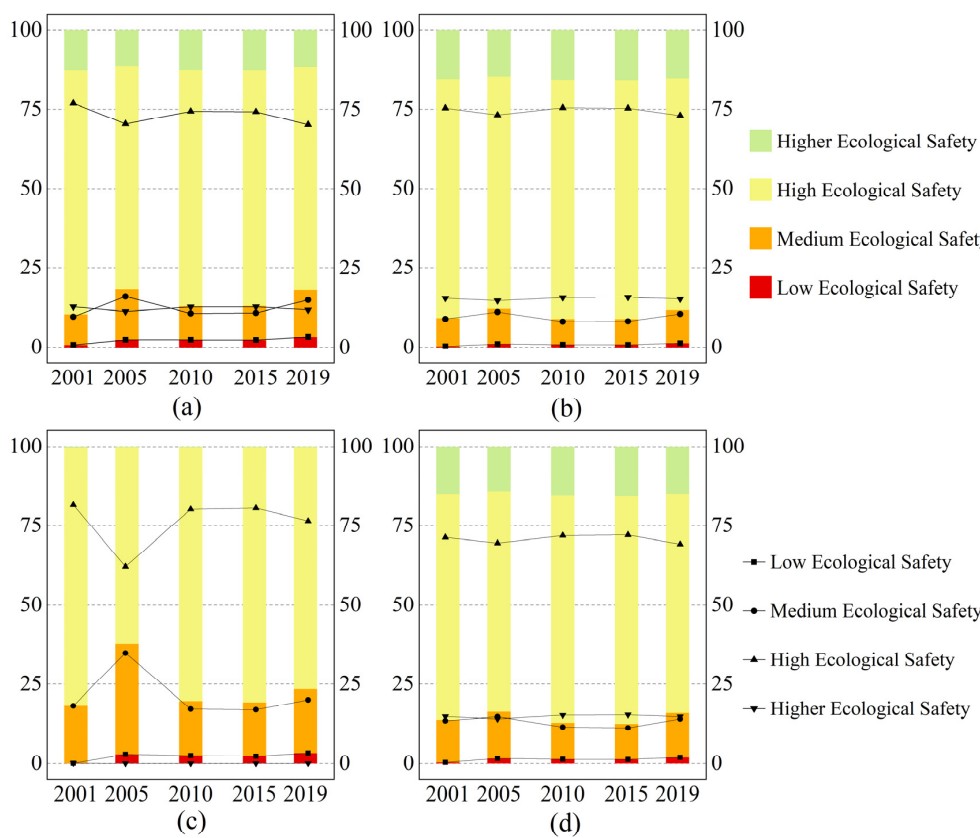

**Figure 5.** Proportions of areas with different ES levels in four provinces: (**a**) Jiangsu, (**b**) Anhui, (**c**) Shandong, and (**d**) Henan.

In Jiangsu, the proportions of areas with low and higher ES did not extensively fluctuate. The proportion of area with low ES reached its maximum value of 3.25% in 2019, and the proportion of area with higher ES reached its maximum value of 12.74% in both 2001 and 2015. The proportions of areas with medium and high ES exhibited opposite change trends. In 2005, the proportion of area with medium ES increased to a maximum of 16.05%, then fell by 6.5%, whereas the proportion of area with high ES decreased by 6.59%.

Between 2010 and 2019, the proportion of area with medium ES continued to increase, whereas the proportion of area with high ES continually decreased until its minimum value of 70.08%.

In Anhui, the proportions of areas for all ES levels did not extensively fluctuate. The proportion of area with low ES was close to 1% throughout 2001–2019. Similar to Jiangsu, the proportion of area with medium ES followed a trend opposite to the trend involving the proportion of area with high ES. In 2005, the proportion of area of medium ES reached its maximum value of 11.07%, then decreased to the minimum value of 7.94% in 2010, and increased again by 2.48% in 2019.

In Shandong, there were only a few areas with higher ES, and the corresponding proportion of area was close to 0. In 2019, the proportion of area with low ES reached its maximum value of 3.36%. The proportions of areas with medium and high ES followed opposite trends. The proportion of area with high ES decreased to its minimum value of 62.14% in 2005, then increased to its maximum value of 80.62% in 2015.

In Henan, the spatial distribution of ES levels fluctuated slightly from 2001 to 2019. The maximum proportion of area with low ES (1.97%) was observed in 2019. The proportion of area with medium ES increased to its maximum value of 14.79% in 2005, then fell to its minimum value of 11.17% in 2015 (decrease of 3.62%). The proportion of area with high ES decreased to 69.54% in 2005, then increased to its maximum value of 72.1% in 2015, and decreased again to its minimum value of 69.21% in 2019 (decrease of 2.71%). The proportion of area with higher ES reached its minimum value of 14.05% in 2005 and its maximum value of 15.40% in 2015.

Overall, the sustainable development ability varied spatiotemporally across the four provinces. On a provincial level, Shandong had worse sustainable development ability, while Anhui had better sustainable development ability. The temporal variations in ES level according to the province were consistent with temporal variations for the whole basin.

### 3.3. Trends in ES value of the HRB

The ES ratings for all grids were calculated each year, and Sen's slope estimator was used to assess trends in the ES status across the basin from 2001 to 2019 (Figure 6). Five trends were distinguished based on the natural points of discontinuity method using ArcGIS: significant decreasing trend, slight decreasing trend, inapparent change, slight increasing trend, and significant increasing trend (Table 4). Then, the proportion of area associated with each trend was calculated for the HRB and Jiangsu, Anhui, Shandong, and Henan provinces.

From 2001 to 2019, 16.33% of the HRB area exhibited an inapparent change trend. These areas were concentrated in the eastern part of the basin. Areas exhibiting significant and slight increasing trends comprised 5.66% and 74.86% of the basin area, respectively. Areas associated with a significant increasing trend were dispersed over the western, southwestern, and southeastern parts of the HRB, whereas areas associated with a slightly increasing trend were distributed across most of the basin. Areas exhibiting significant and slight decreasing trends comprised 0.59% and 2.56% of the total basin area, respectively. When the land use map was overlaid on the trend map, these regions were mainly located in and around built-up areas (Figure 1b).

In Jiangsu, 7.34% of the province's area was associated with a significant increasing trend, more than the proportions in the other provinces. In contrast, 57.23% of the province's area was associated with a slightly increasing trend, the lowest proportion among the four provinces. The proportions of areas associated with inapparent change and decreasing trends were 29.57% and 5.86%, respectively, both higher than the values for the other three provinces. In Anhui, 2.03% of the province's area was associated with decreasing trends, the lowest proportion among the four provinces. Shandong had the lowest proportion of area associated with a significant increasing trend (4.05%). In Henan, areas associated with increasing trends comprised 87.11% of the province's area, the largest proportion of the four provinces.

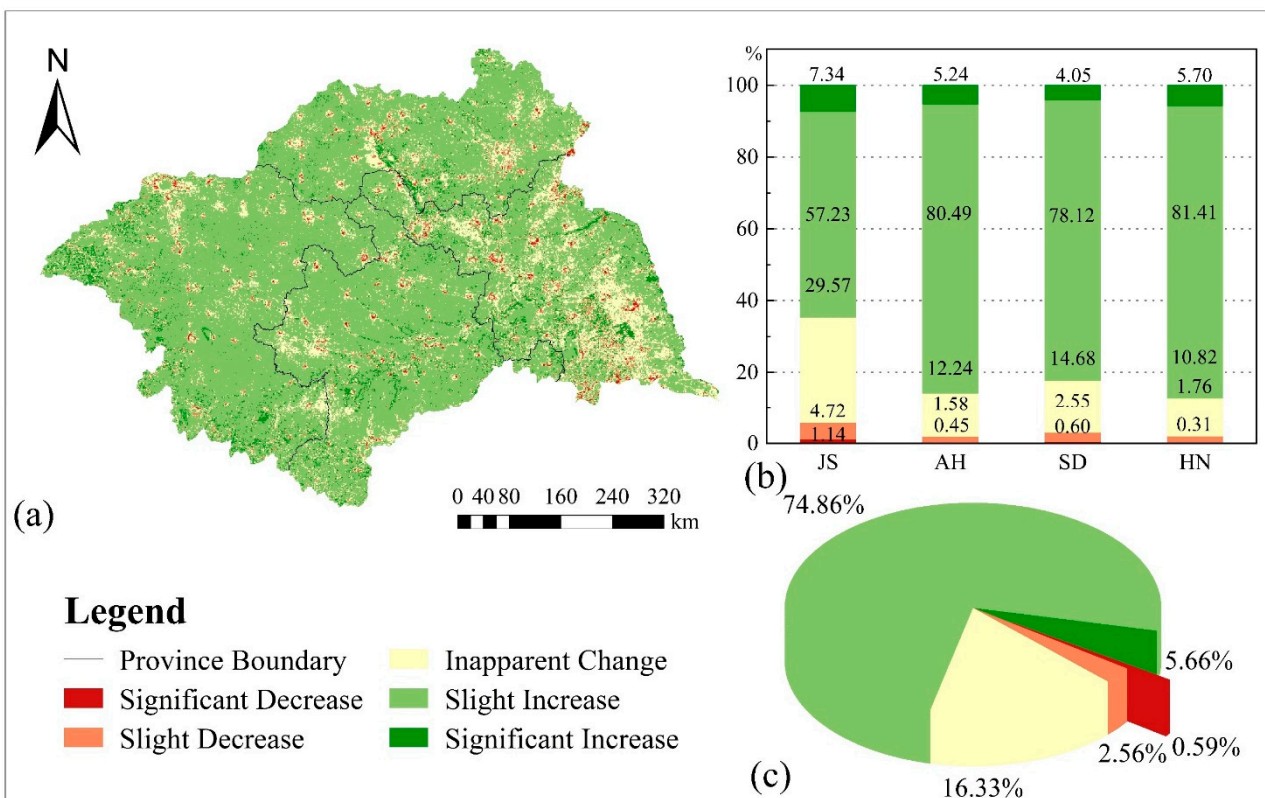

**Figure 6.** (**a**) Change trend of the ES value in the HRB. (**b**) Proportion of area associated with each trend in Jiangsu (JS), Anhui (AH), Shandong (SD), and Henan (HN) provinces. (**c**) Proportion of area associated with each trend in the HRB.

**Table 4.** Classification of the ES variation trend.

| Value | Description |
| --- | --- |
| −0.0191 to −0.0069 | Significant Decrease |
| −0.0069 to −0.0022 | Slight Decrease |
| −0.0022 to 0.0004 | Inapparent Change |
| 0.0004 to 0.0011 | Slight Increase |
| 0.0011 to 0.0124 | Significant Increase |

In conclusion, the ES of the HRB exhibited an improving trend, namely the sustainable development ability of the HRB was getting better. Based on the proportion of area associated with increasing trends, the ES was mostly likely to improve in Henan. The ES variation trend in Jiangsu exhibited the largest regional variation.

## 4. Discussion

The achievement of SDG 15 is essential for human development. There is a lack of models for analyzing the sustainable development of basins. This study established an indicator evaluation system that makes ES analysis more convenient and offers a method to assess sustainable development. By analyzing the spatiotemporal variations in the ES of the HRB, this study contributes to a better understanding of the reasons for eco-environmental degeneration in the HRB, thus facilitating its ecological restoration and achievement of SDG 15.

### 4.1. The Indicator System to Evaluate Sustainable Development of Basins

The HRB has the commonalities of most basins. Industrial and agricultural development and accelerated urbanization have resulted in substantial pressure on the eco-environment in the HRB, and many ecological problems, such as water pollution, land

degradation, and soil erosion, have been reported [59]. Thus, the HRB is well suited to be chosen as the case.

ES refers to the health and integrity of the ecosystem, and is the cornerstone of regional sustainable development [32]. ES assessment is usually carried out by selecting relevant indicators based on a specific framework. Indicators are selected from nature, society, and the economy (e.g., vegetation quality, land-use situation, and distribution of reserves) that inform ES consistent with the United Nations SDGs. Analysis of ES spatiotemporal variation helps to identify issues that need to be prioritized and the causes of ecological problems; thus, it promotes regional sustainable development. Therefore, this study selected ES as the basis for assessing the eco-environmental status of basins. The PSR model is more convenient and applicable than other models [60,61]; thus, this study chose evaluation indices and constructed an ES assessment system based on the PSR framework.

Due to the regional spatiotemporal heterogeneity of ES status, the importance of analyzing differences in spatial distribution is comparable with the importance of analyzing time-series changes. Combining temporal and spatial analyses allows clear and accurate assessment of the evolving regional ecological situation and the causes of such changes. With advancements in spatial information and RS technologies, the assessment of ES can currently include spatial characteristics, such as the combination of various spatial data to analyze changes in ES [26]. Therefore, based on the RS data and GIS techniques, this study determined the ecological situation and its spatiotemporal variations. The HRB covers a wide area, and considering the difficulty of data acquisition and the suitability of temporal and spatial resolution, the study mainly used MODIS data to calculate indicators. Additionally, MODIS products are widely used in ecological research [62] and they are reliable.

Therefore, a comprehensive ES evaluation system based on carefully selected indices from sufficient investigation of the HRB and SDG 15 clearly and precisely reflects the ecological situation and its spatiotemporal variations, hence facilitating follow-up analyses to elucidate the causes of the identified variations.

### 4.2. Analysis of the Factors Influencing Sustainable Development in the HRB

The results showed that the overall ES of the HRB exhibited an improving trend over time. In 2005, the ES level in most areas declined; most of these areas were concentrated in Shandong. However, the best overall ES was attained in 2015 in the HRB. There were obvious differences in the ES level and its change trend among the four provinces in the basin. Overall, the ES was better in Anhui and Henan.

In the 2001–2019 period, the average annual temperature did not extensively change, and ecological factors, such as streamflow and the frequency of natural disasters, did not appear to affect the temperature [63,64], while precipitation has more of an influence on ecological factors. Thus, Sen's slope estimator was used to calculate the change trends of average annual precipitation over 19 years (Figure 7b). The strength of the correlation between the ES level and the average annual precipitation in the HRB was assessed using Pearson correlation coefficients (Figure 7a). The results showed a negative correlation between the ES level and the average annual precipitation in most areas of the HRB, whereas a positive correlation was only observed in a few areas in the southern part of the basin. From 2001 to 2019, the average annual precipitation in most parts of the HRB exhibited a decreasing trend; an increasing trend was only observed in the eastern and southeastern parts of the basin. When these findings were compared with the change trends in the ES level over the same 19 years (Figure 6a), the change trends of ES levels were revealed to be roughly consistent with the change trends of precipitation. Precipitation in almost all areas in the northern, southwestern, and western parts of the HRB exhibited a decreasing trend. In those areas, the ES level was negatively correlated with precipitation; thus, the ES level exhibited an increasing trend. In eastern and southeastern areas, precipitation exhibited an increasing trend and was negatively correlated with the ES level. Therefore, the ES level mainly exhibited a decreasing trend in those areas.

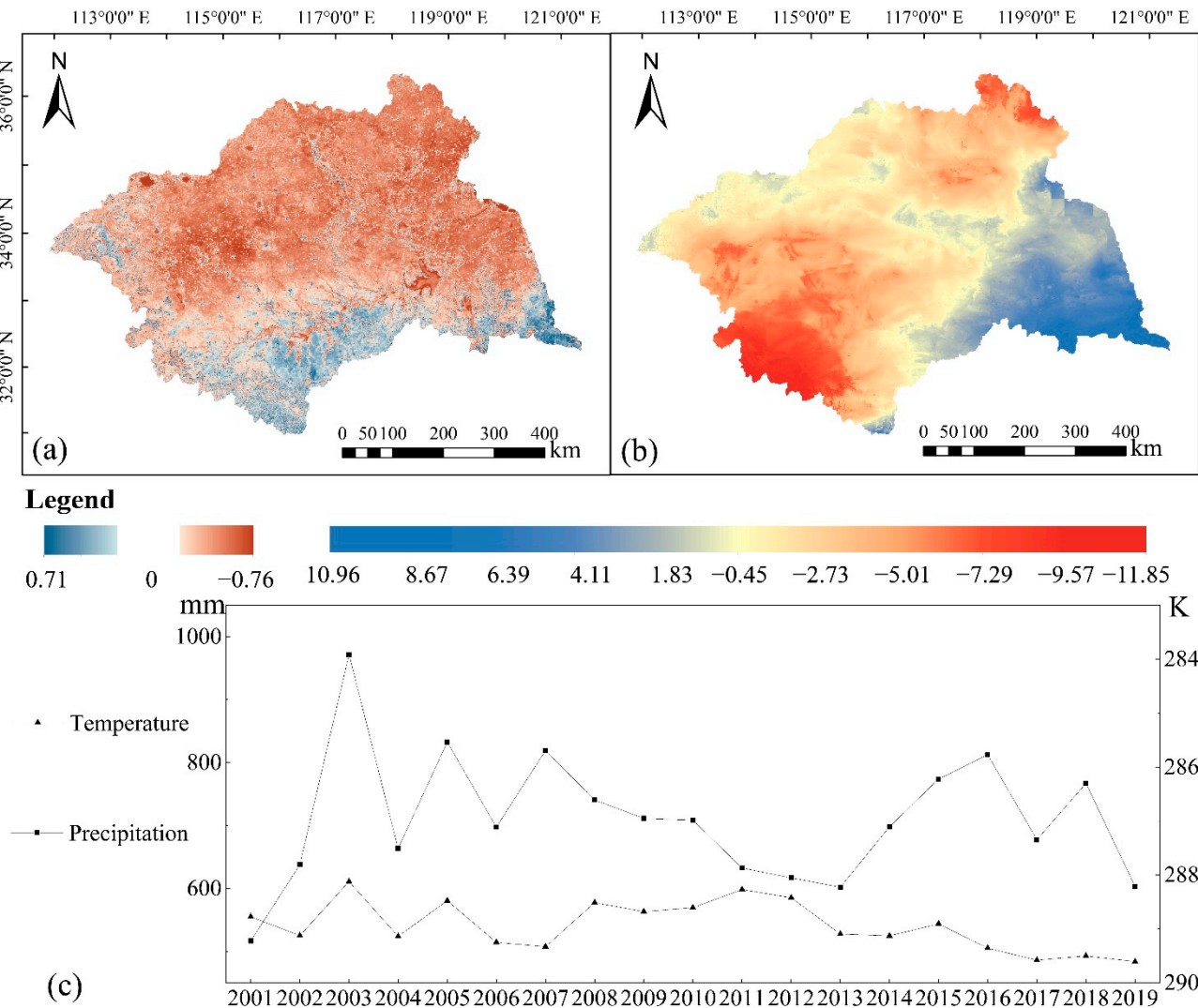

**Figure 7.** (**a**) Spatial distribution map indicating the correlation coefficient between the ES level and the average annual precipitation for each grid. Negative values represent negative correlations, while positive values represent positive correlations; a higher absolute value indicates a stronger correlation. (**b**) Change trends of precipitation in the HRB. Positive values indicate an increasing trend, while negative values indicate a decreasing trend; a lower absolute value indicates a smaller change in precipitation. (**c**) Average annual precipitation and temperature in the HRB from 2001 to 2019.

Using a matrix of land use transfer between 2001 and 2005, spatial distribution maps of the losses (Figure 8a) and gains (Figure 8b) in land-use types between 2001 and 2005 were obtained. Combining these maps with the spatial distribution map of ES levels in 2005 (Figure 3), it is clear that areas in which the ES rating decreased sharply in 2005 were located in Shandong in the northeastern part of the HRB, where a large area of water bodies and wetlands was lost, while the built-up area increased (box 1 in Figure 8a,b). In the eastern HRB, some areas with a decreased ES rating were distributed in a band. Here, grassland was lost and cropland was gained between 2001 and 2005 (i.e., grassland was transformed into cropland; box 2 in Figure 8a,b). These changes in land use types likely caused the decrease in the ES rating for the HRB in 2005. Additionally, from 2001 to 2005, the precipitation level in the HRB was high and fluctuated greatly, which might have adversely impacted the environment.

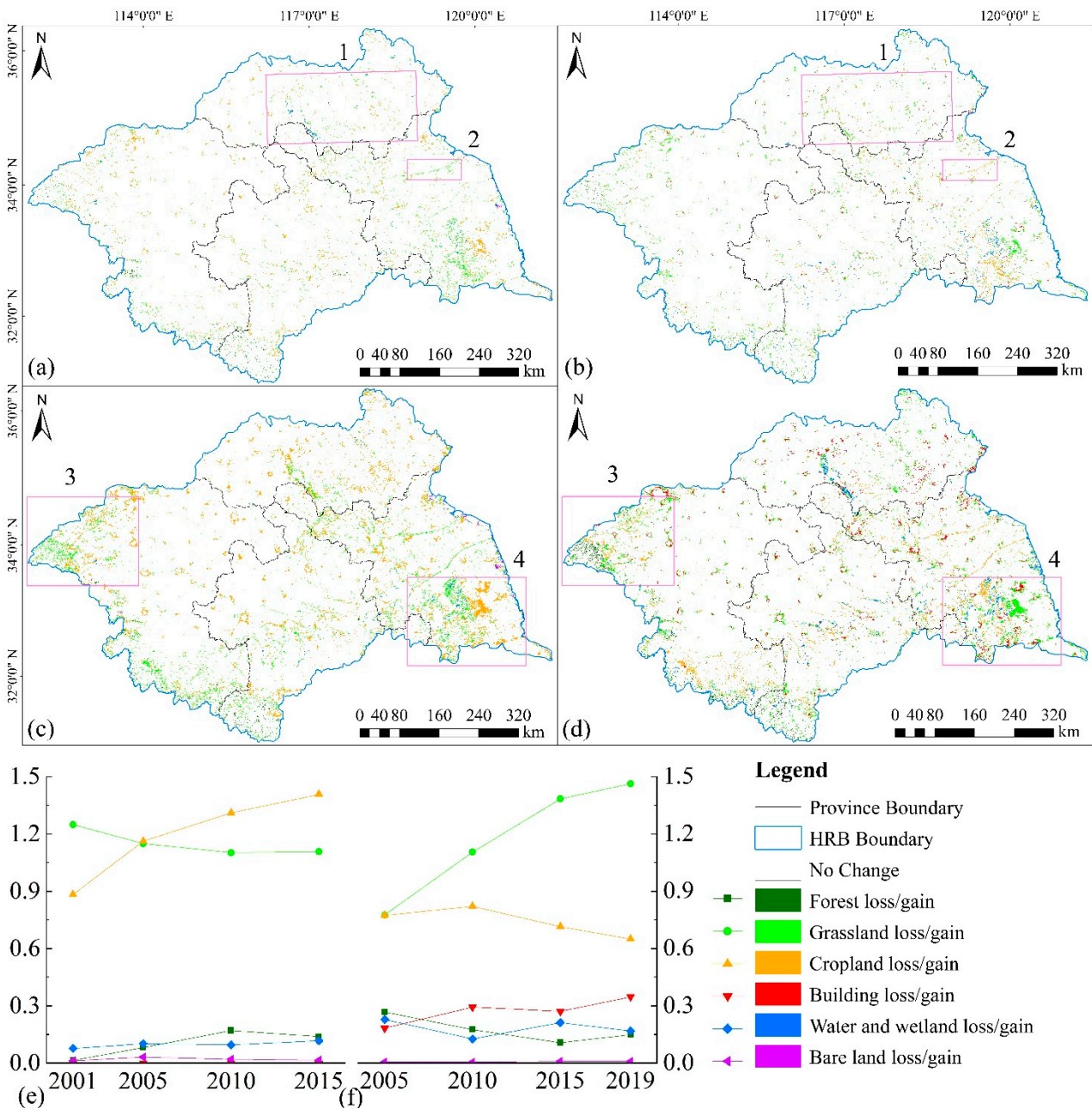

**Figure 8.** Land use transfer maps of the HRB representing (**a**) losses and (**b**) gains in land-use types between 2001 and 2005, and (**c**) losses and (**d**) gains in land-use types between 2001 and 2019. Proportion (**e**) loss and (**f**) gain of each land-use type related to land-use transfer each year.

Based on the change trends of ES (Figure 6a) and the proportion gains/losses in land use types between 2001 and 2019 (Figure 8c,d), the areas exhibiting a decreasing trend in ES roughly corresponded to areas that converted to built-up areas. Therefore, urban expansion is presumably a main cause of environmental degradation in the HRB. The largest increase in built-up area was observed between 2015 and 2019 (Figure 8f), which likely led to the increase in areas with a poor ES status. From 2005 to 2015, the annual precipitation did not extensively change (Figure 7c), the increase in built-up area was small, the area of grassland gained was enhanced, and the area of grassland lost was reduced (Figure 8e). During this period, the environmental status was steadily improving. In 2001–2019, areas exhibiting a significant increase in the ES were concentrated in the western and southeastern parts of the HRB. The land use transfer maps for the 2001–2019

period show that the cropland area in the southeast decreased, while grassland in the same area increased (box 4 in Figure 8c,d). The forest area in the west also increased (box 3 in Figure 8c,d) because of aerial seeding afforestation in Henan [65], thus improving the ES of this region. Furthermore, the waterbody and wetland area increased in 2019, compared with 2001, proving that the "Construction and management of key plains and depressions in the HRB" project was effectively implemented (Figure 9).

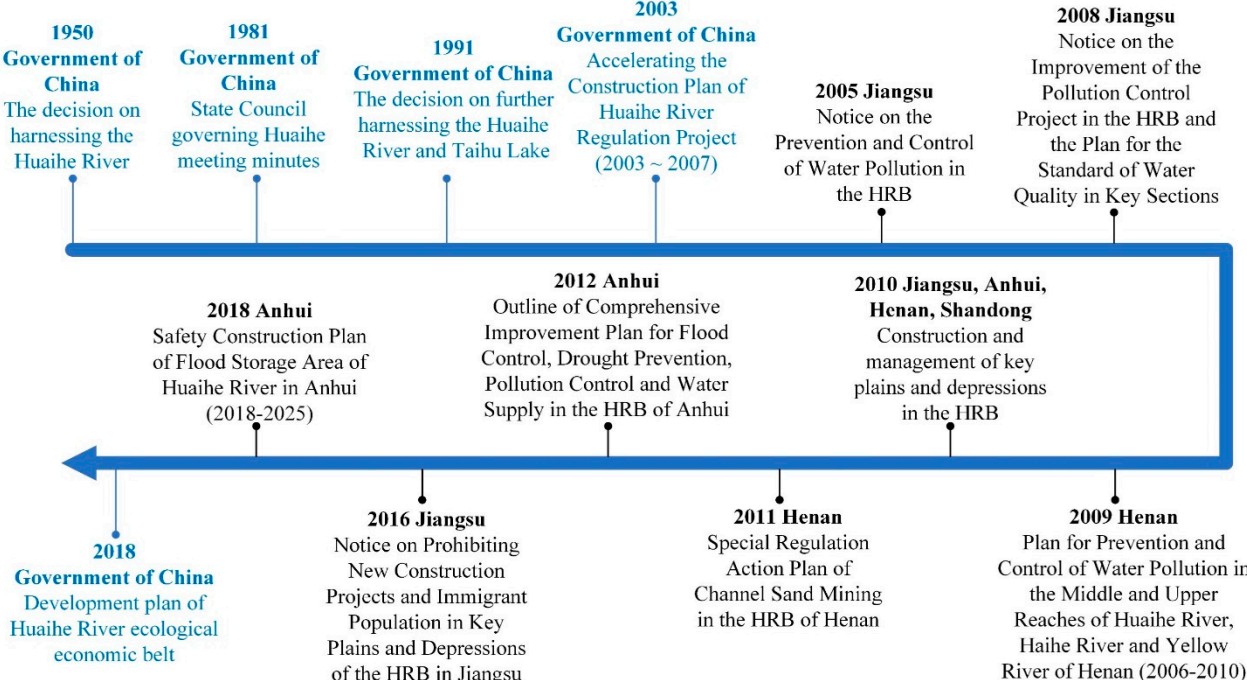

**Figure 9.** Some important policies issued by the Government of China and governments of the four provinces in the HRB.

At the national level, comprehensive management of the HRB has been underway for a few decades (Figure 9). In 1950, the Government of China promulgated its decision to harness the Huaihe River and clearly defined guidelines for regulating the use of the river; the goals were holding back flood waters and ensuring water conservation. By 2018, the Government of China had issued five major documents; the focus had shifted from flood storage to pollution control, ecological restoration, and sustainable development, consistent with changes in socioeconomic development. Each province issued a different number of documents emphasizing various aspects of Huaihe River management between 2001 and 2018 (Figure 9), which may explain the differences in ES status among the four provinces.

### 4.3. Moving toward Achieving SDGs of the HRB

The results of our analyses showed that overall, the ES status of the HRB is good and exhibits an improving trend. The ES of areas that underwent urban expansion declined; thus, although the overall environmental status of the HRB is good, socioeconomic development has exerted negative impacts in specific areas. Areas with improved ecological conditions in the basin experienced increases in forest, grassland, waterbody, and wetland areas, indicating that the measures taken by the government for ecological restoration had positive effects. The afforestation activities in Henan Province have been of help in improving the eco-environment. Simultaneously, in most regions, there was a negative correlation between the change in annual precipitation and the ES value, indicating that the increase in precipitation may destroy the eco-environment.

The Government of China and the provinces within the basin have established governance policies focusing on pollution control and protection against disasters (Figure 9).

Although policies related to these areas are important and should be implemented, government departments should also carefully consider the rational utilization and saving of precipitation and river dredging. It is important to protect and restore land types with high ecosystem value (e.g., forests, grasslands, and wetlands) [66,67], thus improving overall ecosystem function in the HRB.

## 5. Conclusions

This study established a universal ES evaluation system based on the ecological characteristics of the HRB using the PSR model. The indicators are related to the SDG 15, to provide a method to evaluate the sustainable development of basins.

Combining GIS and RS technology, the ES of the HRB was calculated and factors affecting the sustainable development were determined. The results reveal that the spatiotemporal regional distribution of ES differed. Regions near the reserves and buildings are the areas with the best and worst ecological conditions, respectively. The annual precipitation had a negative impact on ES in most areas of the HRB. The addition of grasslands, forests, and wetlands would improve the eco-environment.

The results of these analyses can be used to guide future ecological restoration in the HRB, contributing to the realization of SDG 15. The government needs to focus on water conservancy and vegetation restoration. Some measures, such as aerial afforestation, should be implemented.

**Author Contributions:** Conceptualization, S.S. and T.W.; methodology, S.S.; software, S.S.; validation, T.W., S.W. and S.S.; formal analysis, Y.Y.; investigation, S.S.; resources, T.W. and S.W.; data curation, Y.L.; writing—original draft preparation, S.S.; writing—review and editing, S.S. and T.W.; visualization, S.S.; supervision, M.L.; project administration, Y.Z.; funding acquisition, T.W. and S.S. All authors have read and agreed to the published version of the manuscript.

**Funding:** This research was funded by the Specially-Appointed Professor program of Jiangsu province; National Natural Science Foundation of China, grant number 41671362, 41371359 and the Fundamental Research Funds for the Central Universities, grant number B210203053.

**Institutional Review Board Statement:** Not applicable.

**Informed Consent Statement:** Not applicable.

**Acknowledgments:** Acknowledge the data support from "National Earth System Science Data Center, National Science & Technology Infrastructure of China. (http://www.geodata.cn (accessed on 10 November 2020))".

**Conflicts of Interest:** The authors declare no conflict of interest.

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
