# Peer review of "Ecological Safety Assessment and Analysis of Regional Spatiotemporal Differences Based on Earth Observation Satellite Data in Support of SDGs: The Case of the Huaihe River Basin"

_remotesensing, doi:10.3390/rs13193942_

Round 1

Reviewer 1 Report

The authors have used the PRS model to assess Ecological safety using remote sensing and GIS data. The framework used in this study can be adopted in other areas to evaluate ES and contribute towards SDGs.  

line 16: HRB must be defined

line 166: AHP musts be defined.

What was the consistency ratio after performing AHP? the value must be added in text

Equation 6: specify the data that is used in this equation. 

line 220: specify the grid size as early as possible

Table 3: the authors worked with different spatial resolution from different data sources. How did the authors deal with this problem? this must be described in the paper

line 334-336: Values for Sen's slope categories can be summarized like in Table 2.

Figure 6: Can the authors indicate/describe the significance of the derived trend values

The map scale of figure 1 and others are not consistent

Reviewer 2 Report

The paper establishes a spatiotemporal Environmental Safety evaluation framework that can be used to monitor and assess the realizations of the UN's SDGs 15. One of the important strengths of the approach is its reliance on remote sensing based indicators and on the use of GIS to assess r=the spatiotemporal variation of the ES index. The authors did a great job analyzing spatiotemporal trends and relating them to specific changes in the study area.
The paper is well written, describes fairly well the approach and result analysis. It can benefit from some minor changes:
- The authors selected 5 pressure indicators based on literature. They should discuss what other factors may be considered as pressure factors (energy use, transport for example) and why they were not considered.
- The ES index creation depends on weighted overlay analysis with weights calculated based on AHP. The paper lists the weights in tables but it does not explain how they were derived (how importance was assigned to different layers)
- The trend analysis does not consider inaccuracies of the indicators. What effect does the inaccuray of RS data has on the related indicator and does it compare to variations observed in the trend analysis.\
- In the conclusion, the author indicate that they built a universal ES evaluation system. Can the system be applied in other regions with the same indicators? 

Some minor editing comments:

line 57 : expand PSR when used for the first time
line 121: expand AHP when used for the first time
In figure 2, it is not easy to identify which components the various data sets are used for. Consider using different color lines for the different data sets.
Line 184, the sentence starting 'NPP/VIIRS data' is unclear. Rephrase.
In table 3, can you include the spatial resolution of gridded data (precip and temp). Can you also clarify which temp gridded was used. Is it the 5degx5deg global monthly temp anomaly?
Check section 2.4.4. It seems to have the wrong title.

Reviewer 3 Report

Dear authors, 

Your study seems interesting, but unfortunately, I have to inform you that it needs a plethora of corrections, clarifications, and restructuring. 

To understand the study results, you must address the points I raised in the Pdf attached, then I am more than happy to read the whole manuscript.

Please check all my comments left in the attachment. 

Best wishes, 

Round 2

Reviewer 3 Report

Dear authors, 

Thank you for following my first suggestions and addressed them all. For this recent version, I used track changes in Word where I changed a few words and made a couple of new suggestions. After addressing all of these, I am happy for you to resubmit it and hope to publish it (depending on the other reviewers). 

Best wishes, 
